# Source Identification and Superposition Effect of Heavy Metals (HMs) in Agricultural Soils at a High Geological Background Area of Karst: A Case Study in a Typical Watershed

**DOI:** 10.3390/ijerph191811374

**Published:** 2022-09-09

**Authors:** Qiuye Zhang, Hongyan Liu, Fang Liu, Xianhang Ju, Faustino Dinis, Enjiang Yu, Zhi Yu

**Affiliations:** 1College of Resources and Environmental Engineering, Guizhou University, Guiyang 550025, China; 2College of Agriculture, Guizhou University, Guiyang 550025, China; 3Key Laboratory of Karst Georesources and Environment, Ministry of Education, Guizhou University, Guiyang 550025, China; 4Research and Design Institute of Environmental Science of Guizhou Province, Guiyang 550081, China

**Keywords:** agricultural soils, heavy metals (HMs), source identification, superposition effect, exogenous sources, chemical mass balance (CMB), karst area

## Abstract

Exogenous sources and the superposition effect of HMs in agricultural soils made the idenfication of sources complicated in a karst area. Here, a typical watershed, a research unit of the karst area, was chosen as the study area. The smaller-scale study of watersheds allowed us to obtain more precise results and to guide local pollution control. In this study, sources of HMs in agricultural soil were traced by a CMB model. Superposition effects were studied by spatial analysis of HMs and enrichment factor (EF) and chemical fraction analysis. The average concentrations of Cd, Pb, Cr, Cu, Ni and Zn in surface soils were 8.71, 333, 154, 51.7, 61.5 and 676 mg∙kg^−1^, respectively, which exceeded their corresponding background values. The main sources of Cd, Pb and Zn in agricultural soil were rock weathering, atmospheric deposition and livestock manure, and their contributions were 47.7%, 31.0% and 21.2% for Cd; 7.63%, 78.7% and 13.4% for Pb; and 17.0%, 52.3% and 28.1% for Zn. Cr mainly derived from atmospheric deposition (73.8%) and rock weathering (20.0%). Cu and Ni mainly came from livestock manure (81.3%) and weathering (87.5%), respectively, whereas contributions of pesticides and fertilizers were relatively limited (no more than 1.04%). Cd, Pb, Zn and Cu were easily enriched in surface soils near the surrounding pollution sources, whereas Cr and Ni were easily enriched in the high-terrain area, where there was less of an impact of anthropogenic activities. The superposition of exogenous sources caused accumulation of Cd, Pb and Zn in topsoil, contaminated the subsoil through leaching and improved bioavailability of Cd and Pb, causing high ecological risk for agricultural production. Therefore, Cd and Pb should be paid more attention in future pollution control.

## 1. Introduction

Heavy metal (HMs), enrichment in the human body through the food chain has been recognized as one of the most important constituents of global environmental pollutants that are considered to be toxic to human and biota [1,2,3]. For example, Pb causes harm to the nervous, enzymatic, endocrine and immune systems; Cd is responsible for lung cancer, kidney dysfunction, nephrotoxicity and pulmonotoxicity [4]. Crops mainly uptake HMs from agricultural soils, and a high content of HMs in soils can contaminate food as well as decrease crop productivity, causing adverse physiological changes that may even lead to cell death. Thus, source identification and the superposition effect of HMs in agricultural soils are the basis of pollution control and ensuring food security.

Guizhou Province is the centre of the karst region in southwest China, which is widely covered with carbonate rock. As Cd replaced Ca in calcite at the process of deposition [5], the background of Cd was higher than in the non-karst region. According to a China National Environmental Monitoring Center investigation (1990), the background value of Cd was 0.66 mg∙kg^−1^ [6]. A research study about the background values of elements in the surface soils of China also found that concentrations of Cd and Zn in the surface soil of Guizhou were 0.752 and 138.9 mg∙kg^−1^ [7], which were almost higher than in other provinces of China. As a result of wide distribution of HMs in this area, additionally, the background levels of Cd, Pb and Zn were higher than in other regions. The northwest region of Guizhou, a typical karst area, distributed a lot of lead and zinc deposits and had a long history of zinc smelting by indigenous methods (over 300 years) [8]. In the 1980s, especially, many zinc smelters caused serious soil pollution with Cd and Pb. Furthermore, a lot of small dispersive zinc smelting slag, containing a high content of HMs [9], can be seen everywhere in this area. This slag polluted surrounding soils through surface runoff and dust fall and migrated to farmland [10,11,12]. Therefore, this area has been paid more attention in recent years [13].

Accumulation of heavy metals (HMs) in agricultural soil has posed a great threat to agricultural production in recent decades. The superposition of exogenous sources was most significant for soil pollution. Exogenous sources of HMs included mining, industrial sources (such as the metal smelting plants, thermal power plants and alkylation plants) [14,15,16,17], agricultural sources (fertilizers, pesticides and livestock manure) [18,19] and traffic sources (wear of metal parts and exhaust emission) [20]. In addition, the soil of the karst area, compared to that of non-karst area, is characterized by thin regoliths, uneven distribution, and high porosity [21,22], which contributed to the migration of HMs from the surface to the subsoil and even contaminated underground water [9,23]. The process by which HMs entered crops from soils consisted of three stages: the movement of HMs in soils, accumulation of HMs in crop roots and transportation of them to the shoots [24]. Thus, chemical fractions were indicators of the mobility of HMs and were more meaningful than content with regard to HMs’ biotoxicity [25,26]. HMs in the soils formed different fractions with ion groups in the soil through various physical and chemical processes [27], which were related to soil type, soil property, exogenous sources and environmental conditions [28]. Furthermore, the roots of plants also played an important role in the increase of HMs’ bioavailability for generation of some compounds such as sugars, organic acids and amino acids in the root zone [29]. Therefore, the pollution from exogenous sources not only increased the content of HMs in the soil but also probably enhanced their mobility, thus generating high ecological risk. Generally, source control was the most effective method to reduce HMs pollution. However, it was because of the impact of multiple sources that the sources of HMs were difficult to identify in agricultural soil.

Currently, numerous methods were utilized in source apportionment of HMs in soils. For instance, correlation analysis and cluster analysis were widely used to qualitatively define these sources [30,31], whereas receptor models, such as principle component analysis (PCA), chemical mass balance (CMB), positive matrix factorization (PMF), UNMIX and absolute principal component analysis followed by multiple linear regression (APCA-MLR), were widely applied to quantitatively analyze the contribution of each source [14,32,33,34]. Among the mentioned methods, the CMB model, directly analyzing numerical values rather than deviations, had a better-fitting accuracy based on building of the source profile, which improved confidence in the robustness of source apportionment results [4].

The existence of many peaks and rivers in the karst area divided the area into many relatively independent, small watersheds, most of which had unique charactersitics, such as geology, industrial enterprises and historical pollution. Hence, source identification and the study of accumulation of HMs in agricultural soils at a karst area should start from a small watershed. Moreover, previous studies on soil pollution of the area merely focused on the content of HMs and ecological risk and roughly attributed the main source of HMs to industrial activities [21,35,36], which were mostly generalized, whereas the study on the smaller-scale watershed made for obtaining more precise results and better guiding of local pollution control.

In this study, a typical watershed in the northwest of Guizhou Province was chosen as the study site. The objectives of the study were to trace sources of HMs of agricultural soils by using CMB and to figure out the superposition effect through spatial distribution and chemical fraction analysis of HMs in agricultural soils. To that end, source identification was conducted by chemical mass balance (CMB) based on the monitoring of seven potential sources (irrigation water, livestock manure, pesticides, wet deposition, dry deposition and rock). The enrichment degree of HMs was described by enrichment factor (EF) and the chemical fraction of HMs was analyzed by the method of Community Bureau of Reference (BCR). The framework of this study is shown in Figure 1.

## 2. Materials and Methods

### 2.1. Study Site

The study site (centre coordinates: 104.384635 E, 26.78406 W), a typical watershed between two mountains, is located in Maoshui Village, Weining County (Figure 2). This area is a subtropical humid monsoon climate zone, and its average rainfall and temperature were 926 mm and 18 °C, respectively. Emergence stratums were Huanglong formation of Upper Carboniferous and Baizuo, Shangsi and Xiasi formation of Lower Carboniferous, indicating that this area is covered with carbonate rocks. Additionally, two pollution sources, a zinc smelting plant and the slag of zinc smelting, are distributed in this area, as shown in Figure 2. The main irrigation water for adjacent farmland (dry-land) is a small stream (Maoshui River) flowing along the middle of the study area.

### 2.2. Sample Collection and Chemical Analysis

#### 2.2.1. Sample Collection

In this study, 45 surface soil samples were collected from local farmland in April 2021 (Figure 2, red points). In order to assess the impact of industrial sources, 62 detailed samples of surface soil were collected at the nearby zinc smelting plant (Figure 2, yellow points). Each surface soil sample comprised 5 samples, which were mixed vigorously and cleaned of gravel and plant residues. In addition, 5 soil profiles, from a depth of 120 cm, were studied here. A total of 6 soil samples were collected from every profile, meaning once in every 20 cm from the surface to the 120 cm. All collected samples were sealed in polyethylene bags and transported to the laboratory and air-dried at room temperature. For a better confirmation about potential sources of HMs in soil, a detailed investigation was conducted before sample collection. According to the survey results, potential sources included rock weathering, industrial sources (the zinc smelting plant and the slag of zinc smelting) and agricultural sources (fertilizers, pesticides, livestock manure and irrigation water). Therefore, 7 rock samples (collected from surrounding mountains), 7 irrigation water samples (collected from Maoshui River and surrounding ditches), 6 livestock manure samples (collected from farmland and farmer’s home), 6 pesticide samples and 2 fertilizer samples (collected from local dealers) were collected. Moreover, 5 sample sites of atmospheric deposition (Height of deposition cylinder: 5~15 m) were set here (Figure 2), and samples of dry deposition and wet deposition from May 2021 to July 2021 were collected, once a month, for a total of 15 samples.

#### 2.2.2. Sample Processing

(1)Soil samples

All soil samples were finely ground in a porcelain mortar and passed through a 100-mesh polyethylene sieve and stored in a desiccator before analysis. The total concentration of the elements in the soil sample was digested with a mixture of HNO_3_-HF-HClO_4_ (5:1:1) in a closed poly-tetrafluoroethylene system at 180 °C for 10 h. Then, the soil sample was put on a 140 °C hot plate and heated to about 1 mL, and it was immediately transferred to a 25 mL volumetric tube. The inner wall of the closed poly-tetrafluoroethylene was rinsed with deionized water and also transferred to the tube to dilute to the tick mark. After that, the sample was filtered through a 0.45 μm cellulose acetate filter membrane before measurement.

The sequential extraction procedure of chemical composition of HMs was complied by the Community Bureau of Reference (BCR), and the specific operation refers to the study of Yebpella [37]:(2)Livestock manure samples

All livestock manure samples were dried at room temperature and broken by the stainless-steel crusher (FW100). Then, these samples were digested with a mixture of HCl-H_2_O_2_ (GR, volume ratio 3:1) in a closed poly-tetrafluoroethylene system at 180 °C for 10 h. The subsequent procedures were same as those for the treatment of the soil sample.

(3)Irrigation water samples

10 mL of irrigation water (liquor) was added in a 50 mL beaker that was heated and evaporated to dryness on a hot plate. A total of 12.0 mL HNO_3_ was added and continuously heated until the color of liquor became clearly pale. Then, 3.0 mL of HClO_4_ was added in the beaker, which was heated until the emission of white smoke. Then, the beaker was removed from the hot plate and cooled to room temperature. The rest of the liquor was diluted with distilled water to volume (13 mL) and mixed. Finally, the obtained liquor was filtered by filter membrane (0.45 μm) before measurement.

(4)Rock samples

Rock samples were finely ground. Next, the treatments followed the same procedures as those used to process the soil samples. Finally, the samples were measured after being filtered through a 0.45 μm cellulose acetate filter membrane.

(5)Atmospheric deposition samples

Every atmospheric deposition sample was divided into two parts (dry deposition and wet deposition). Impurities, such as branches, leaves and insects, were picked out of the settlement urn and the supernatant, filtered by three layers of filter paper, was collected as a wet deposition. Next, dust particles (the solid phase) were separated and evaporated to dryness in a porcelain crucible. Then, these dust particles were dried further to constant weight in an air oven (105 ± 5 °C), and a dry deposition sample was obtained for determining the concentration of HMs. Next, treatment of the wet deposition and dry deposition was similar to that of the irrigation water samples and soil samples, respectively.

#### 2.2.3. Chemical Analysis

All pretreatment solutions were used to determine the concentration of Cd, Pb, Cr, Cu, Ni, Zn and Al by inductively coupled plasma mass spectrometry (ICP-MS) (Agilent 7900, Agilent Technologies (US)). All experimental containers were cleaned and immersed in 10% nitric acid solution for over 24 h.

### 2.3. Chemical Mass Balance (CMB)

Chemical mass balance (CMB) was widely applied in source apportionment. This model calculated source contributions based on the following hypotheses: (1) chemical composition of pollutants exhausted from all sources were different and relatively stable in the transport; (2) there was no existence of interdependence among all pollutants emitted from all sources; (3) the source contribution of each source was definite; (4) the source profiles of sources were relatively independent and non-collinear. According to CMB calculation, the pollutant concentration of the receptor point was equal to the linear addition of all sources’ pollutants, which can be expressed as follows:(1)$$c_{i}=j=pF_{ij}S_{j}(i=1,\ldots ,m)$$
where $c_{i}$ is test concentration of *i*th heavy metal, mg∙kg^−1^; *p* is the number of sources (*j* = 1, 2, 3,..., *p*); $F_{ij}$ is concentration of *i* the heavy metal in *j* the source, mg∙kg^−1^; $S_{j}$ is the concentration of each source’s contribution and *m* is the number of the heavy metal (*i* = 1, 2, 3,..., *m*). Hence, the contribution of *j* the sources ($\eta_{i}$) can be calculated as following:(2)$$\eta_{i}=S_{j}c\times 100\%$$
where, *c* is total concentration, mg∙kg^−1^.

In this study, the whole explored area was regarded as a receptor point. Hence, average values of Cd, Pb, Cr, Cu, Ni and Zn were utilized in calculation.

### 2.4. Superposition Effect of Exogenous HMs

In this study, the superposition effects of exogenous HMs in agricultural soils were studied by spatial distribution and chemical fractions. Horizontal distribution can help to identify potential pollution sources through spatial variation analysis. In order to further figure out the impact of anthropogenic sources, the enrichment factor was used to describe accumulation of HMs (Cd, Pb, Cr, Cu, Ni and Zn) in surface soils and soil profiles.

The enrichment factor is an important parameter for assessment of the anthropogenic impact on soil, and it can be expressed as follows [38]:(3)$$EF=\frac{{\left(C_{i}/C_{n}\right)}_{soil}}{{\left(C_{i}/C_{n}\right)}_{background}}$$

$C_{i}$ denotes concentration of HMs (Cd, Pb, Cr, Cu, Ni and Zn) in soil, mg∙kg^−1^; $C_{n}$ denotes standardized elements (Al, Fe and Ti, etc.) [39,40], mg∙kg^−1^. In this study, Al is chosen as the standardized element. Backgrounds of Cd, Pb, Cr, Cu, Ni, Zn and Al refer to “Chinese Soil Element Background Value” [6]. According to *EF*, as shown in Table 1**,** pollution was divided into six levels [41].

Based on the result of spatial distribution analysis, the influence of exogenous sources on chemical fractions of agricultural soils was studied by comparing the differences of chemical fractions between high-impact regions and lower-impact regions.

### 2.5. Data Analysis

All data preprocessing and transformation were processed using Microsoft Excel 2007. The T-test was used to check the outliers of all experimental data, and the Kolmogorov–Smirnov (K–S) test was used to test the normality of the data. SPSS statistic 25.0 was used for the analysis of variance (ANOVA) test, T-test and K–S test. All data were rounded off to three significant figures in the study. The analysis of chemical mass balance (CMB) was performed by EPA CMB 8.2. Origin pro 8.0 was used to draw histograms of the chemical fraction of HMs. Semivariance analysis of total content of HMs of surface soils was conducted through GS + 9.0. Spatial distribution of HMs and the corresponding EF were conducted by the Kriging toolbox in ArcMap 10.2.

## 3. Results and Discussion

### 3.1. Concentration of HMs

The basic statistical information of the selected HMs—Cd, Cr, Cu, Ni, Pb and Zn—are shown in Table 2. The results revealed that the concentration range of the HMs in the soil are Cd (2.12–24.5 mg∙kg^−1^), Pb (76.0–1530 mg∙kg^−1^), Cr (105–286 mg∙kg^−1^), Cu (24.1–131 mg∙kg^−1^), Ni (36.2–104 mg∙kg^−1^) and Zn (210 to 3340 mg∙kg^−1^). Their mean values were 8.71, 333, 154, 51.7, 61.5 and 676 mg∙kg^−1^, respectively, revealing that the values exceeded their corresponding background values of topsoil in Guizhou Province (Table 2). For the estimated coefficient of variation (CV), Cr (25.9%) and Ni (28.5%) showed low variability, and Cd (40.3%) and Cu (42.7%) showed moderate variability; by contrast, Pb (77.2%) and Zn (77.9%) presented high variability. This indicated that Cr and Ni were less affected by anthropogenic activities. By contrast, Cd, Cu, Pb and Zn revealed superposition of exogenous sources to varying degrees in the surface soil [42,43], especially Pb and Zn.

### 3.2. Source Identification of HMs

#### 3.2.1. Contributions of Sources

The monitoring results of seven sources are presented in Table 3. The results indicate that all concentrations of Cd, Pb, Cr, Cu and Zn in livestock manure and atmospheric deposition were higher than those in other sources, implying the two sources probably were main sources of Cd, Pb, Cr, Cu and Zn. In addition, the content of Ni in rock was higher than in other sources except for dry deposition, so rock weathering may be one of the main sources of Ni.

The CMB calculations are shown in Table 4. It can be seen that R^2^ and χ^2^ are 1 and 0, respectively. Moreover, all values of T_stat_, the standard deviation ratio of contributions of two sources, are greater than 2. Therefore, the fitting degree of the calculation was high, signifying that the result of CMB was reliable. The contribution of every source for Cd, Pb, Cr, Cu and Zn is depicted in Figure 3. It is clear that the impact of irrigation water on HMs was limited and could be neglected in pollution sources control. The main sources of HMs were livestock manure, pesticide, wet deposition, dry deposition, rock weathering and fertilizer; their contributions were 21.2%, 0%, 17.5%, 13.4%, 47.7% and 0.190% for Cd; 13.4%, 0.100%, 2.48%, 76.2%, 7.63% and 0.100% for Pb; 3.70%, 1.00%, 68.2%, 5.60%, 20.0%, and 1.50% for Cr; 81.3%, 0, 0, 15.1%, 2.60% and 1.00% for Cu; 5.70%, 0.100%, 0, 5.90%, and 87.5% for Ni; and 28.1%, 0.570%, 24.4%, 27.9%, 18.0% and 1.04% for Zn. Although pesticides and fertilizers were widely used for agricultural production, their contributions to HMs were limited (no more than 1.04%).

#### 3.2.2. Process of Contamination Superposition

The source contribution analysis showed that rock weathering was the dominant source for Ni and Cd, indicating that the influence of exogenous sources on Ni and Cd was limited. Controlling the effect of the high geological background in the carbonate rock area was the influence of the high content of Cd in the agricultural soils. Livestock manure such as cow dung, swine manure and goat manure have a long history as a great organic fertilizer in Chinese rural areas; this fertilizer causes accumulation of HMs in agricultural soils [44]. Compared to other HMs, Cu (II) and Pb (II) can effectively bond to dissolve organic matter (DOM) in livestock manures [45]. Some studies also indicated that long-term application of livestock manure generated accumulation of Cu in soils [46,47,48]. Hence, the main source of Cu in agricultural soils was livestock manure in this area.

In addition, atmospheric deposition was a significant source for Cd, Pb, Cr and Zn. Because of the presence of a zinc smelting plant and smelting slag in this area, contributions of atmospheric deposition for Cd, Pb, Zn and Cr were relatively high [17]. Zinc smelter primarily polluted surrounding soil through atmospheric deposition, in which HMs mainly came from flue gas emissions and ground dust. Bi et al. estimated that approximately 450 t of Cd was released into the local atmosphere from zinc smelters at Hezhang County from 1989 to 2001 [49]. Despite the fact that indigenous zinc smelting had been forbidden, the long smelting history has contaminated surrounding soils, which generated concentrations of HMs in this area that were higher than they were in others [9,50]. With the help of wind, soils also became an important source of atmospheric deposition [51,52]. Nevertheless, it can be found that contributions of wet deposition and dry deposition appeared to be different for diverse HMs, which was related to their state of existence and chemical fractions in the atmosphere. Some HMs were adsorbed on atmospheric particulates, whereas others dissolved in water or reacted with acid (sulfuric acid and nitric acid), which would have transferred HMs from a solid phase to a liquid phase and formed droplets or aerosol in the atmosphere. The Pb fraction in the dust of the zinc smelting plant principally included oxide, sulfate, elementary substance and arsenate [53], and they were not dissolved in water. In addition, Pb in the slag existed in lead sulfate, and the elementary substance was adsorbed onto other minerals or covered with vitreous [50,54] matter. Therefore, Pb in the atmosphere mainly existed in a solid phase, whatever the entry mode, causing dry deposition as the main contributor in soil. Cr, which mainly exists in organic and residual form in coal, was released through coal combustion and generated sulfate with sulfur dioxide [55,56], which displayed solubility in water. Therefore, wet deposition became a main contributor for Cr, Cd and Zn in soot containing sulfate and oxide, whereas the existing forms in the slag were complex, and most of them were enfolded with vitreous [54] matter. Hence, the contributions of wet deposition and dry deposition were similar to Cd and Zn.

### 3.3. Superposition Effect of HMs

#### 3.3.1. Spatial Distribution of HMs

1.Horizontal distribution

Spatial variation analysis of HMs was conducted in order to determine the influence range of exogenous sources. Semivariance was the key to spatial variation analysis. It reflected the spatial variability structure of the regionalized variable and played a significant role in Kriging interpolation [13]. The results of the semivariance analysis and correlation parameters are presented in Table 5. According to the principle of the maximum of determination coefficient (R^2^) and minimum of residual square sum (RSS), the best fitting modes for Cd, Pb, Cr, Cu, Ni and Zn were selected. Because all values of R^2^ were beyond 0.6, spatial fitting models for HMs were reasonable and acceptable. Nugget is random variation; sill is the maximum variation between data pairs; the ratio of nugget to sill (C_0_/C_0_ + C), the proportion of random variation in sum variation, can be used to express spatial autocorrelation of regional variables and reflect the predominant factor. The values of C_0_/C_0_ + C for Cd, Pb, Cr, Cu, Ni and Zn were below 25%, indicating a strong spatial correlation in the corresponding range, which explained that these HMs were primarily attributable to a natural factor (such as parent material, soil type, topography, etc.) or atmospheric deposition. Moreover, the higher values of C_0_/C_0_ + C for Cd and Pb demonstrated more anthropogenic impact.

Based on semivariance analysis, horizontal distributions of HMs were drawn, as shown in Figure 4. The concentrations of Cd, Pb and Zn were correlated to distance from the smelting plant and the smelting slag, and high contents were discovered at the centre and in the vicinity of the zinc smelting plant and in the zinc smelting slag. In the prevailing wind direction—the south–north direction—the contents of Cd, Pb and Zn gradually decreased with the increase of distance from the plant, proving that industrial activities have polluted surrounding agricultural soils. However, the distribution character of Cu was different, implying that the main source of Cu in soil was not industrial sources. According to a previous study (Section 3.2.1), livestock manure contained a lot of Cu, and many composts of livestock manure were used in farmland, which further proved livestock manure was the main source of Cu. In terms of Ni and Cr, their distribution characteristics were in keeping with elevation, which demonstrated Cr and Ni were mainly due to rock weathering or atmospheric deposition [57]. Also, it can be found that high contents of Ni and Cr appeared at high elevation areas as a result of the terrain’s inhibition of the movement of ground dust. On the other hand, Cr was not exactly same as Ni: its highest value was close to the west, near local residents, indicating that coal combustion from nearby residents probably was one of the sources for Cr.

Judging by the EF of Cd, Pb, Cr, Cu, Ni and Zn in the surface soil (Figure 5), the Cd in surface soils has reached significant enrichment. Pb and Zn were also between moderate enrichment and significant enrichment, indicating that Cd, Pb and Zn have become significantly enriched in surface soils due to the influence of exogenous sources. However, the lower EF of Cr and Ni indicated less influence of anthropogenic activities. Therefore, Cd, Pb, Zn and Cu were easily enriched in the areas surrounding the pollution sources, whereas Cr and Ni easily enriched the high-terrain area.

2.Soil profiles

The EF of HMs (Cd, Pb, Cr, Cu, Ni and Zn) in the soil profiles is described in Figure 6. The EFs of Cd, Pb and Zn in the topsoil were clearly higher than those in the subsoil, and with an increase of depth, the EF showed a general downward trend, indicating that the accumulation of Cd, Pb and Zn in the topsoil is due to exogenous sources. By contrast, the EF of Cr, Cu and Ni profiles remained almost unchanged, indicating less impact of exogenous sources. Moreover, the EFs of Cd, Pb and Zn in the topsoil were higher than those of others, indicating they were influenced by exogenous sources. The higher EFs of HMs were in the topsoil; they were also higher in higher parts of the subsoil (such as SF-4, SF-5 in Figure 6b–f), implying that the subsoil of the study area has been polluted through vertical migration.

#### 3.3.2. Chemical Fraction of HMs

Since the main influence area was located in the centre of the study area, near the zinc smelting plant and the slag, 62 detailed soil samples were collected in the region, and the chemical fractions of Cd, Pb and Zn were analyzed for more severe contamination. The results (Figure 7) obtained showed that the distribution of individual HMs in surface soils, on average, followed the same pattern as the Cd presented: Reducible > Exchangeable > Residual > Oxidizable; Pb: Reducible >Residual > Oxidizable > Exchangeable; Zn: Oxidizable > Residual > Exchangeable > Reducible. Obviously, all non-residual fractions of Cd, Pb, and Zn were greater than their residual fractions. Regarding Zn, the high proportion of oxidizable fraction (42.3%) and residual fraction (38.7%) implied low ecological risk. Moreover, the high content of exchangeable fraction (1.48 mg∙kg^−1^) and reducible fraction (1.94 mg∙kg^−1^) explained the high bioavailability of Cd. Although the exchangeable fraction of Pb was low, the content of reducible fraction (121 mg∙kg^−1^) was high, accounting for 62.7% of the total content, indicating high potential ecological risk. This is because the reducible fraction easily transformed to an exchangeable fraction due to the influence of the external environment, which also applied to Cd. Therefore, the superposition of exogenous sources increased the contents of Cd and Pb simultaneously. Their migration in the soil increased their exchangeable and reducible fractions, causing a great ecological risk for local agricultural production.

## 4. Conclusions

In this study, the sources of HMs and the superposition effect of HMs were investigated. The main sources of Cd, Pb and Zn in agricultural soil were rock weathering, atmospheric deposition and livestock manure, and their contributions were 47.7%, 31.0% and 21.2% for Cd; 7.63%, 78.7% and 13.4% for Pb; and 17.0%, 52.3% and 28.1% for Zn. Cr mainly derived from atmospheric deposition (73.8%) and rock weathering (20.0%). Cu and Ni mainly came from livestock manure (81.3%) and weathering (87.5%), respectively, whereas contributions of pesticides and fertilizers were relatively limited (no more than 1.04%). Moreover, Cd, Pb and Zn were greatly influenced by the zinc smelter and zinc smelting slag. The superposition of exogenous sources caused accumulation of Cd, Pb and Zn in surface soils, contaminated the subsoil through leaching and improved bioavailability of Cd and Pb, causing high ecological risk for agricultural production. Therefore, Cd and Pb should be paid more attention in future pollution control.

## Figures and Tables

**Figure 1 ijerph-19-11374-f001:**
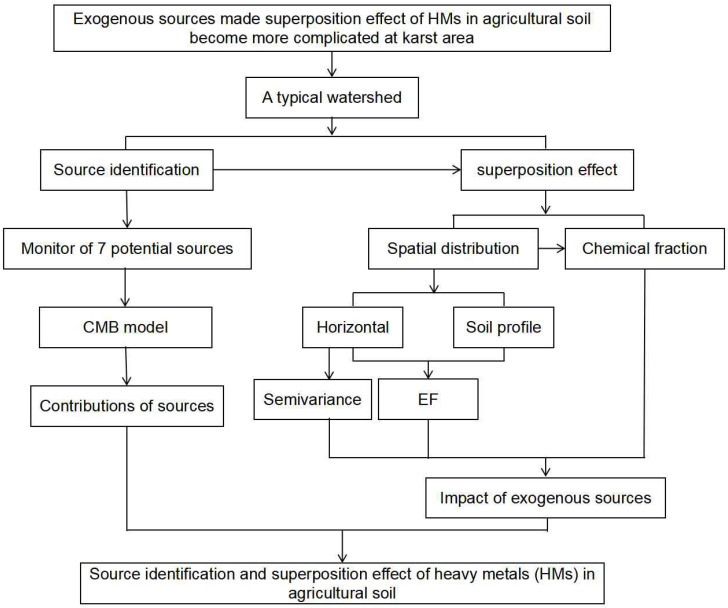
The framework for study of source identification and superposition effect of HMs.

**Figure 2 ijerph-19-11374-f002:**
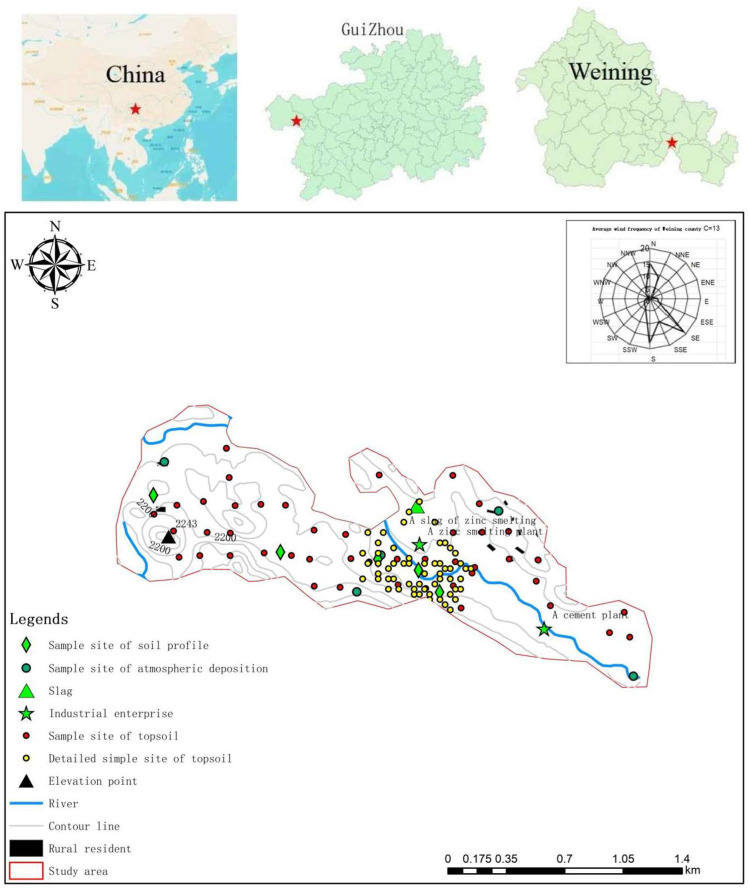
Study area and sample sites.

**Figure 3 ijerph-19-11374-f003:**
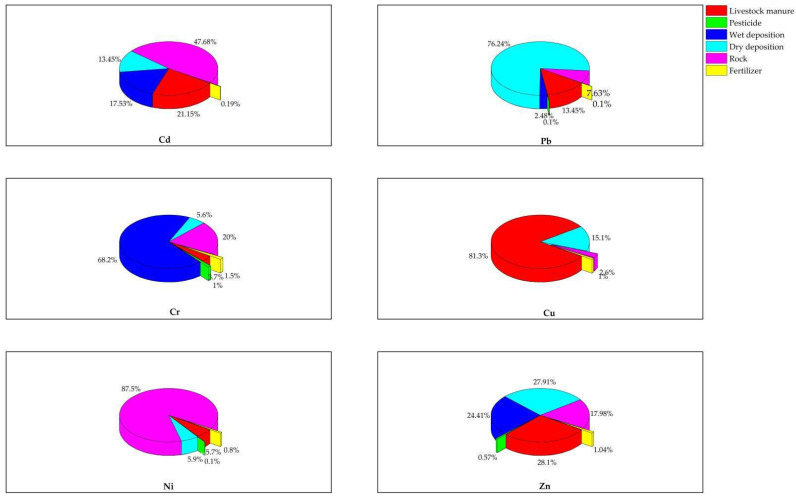
Contributions of pollution sources for Cd, Pb, Cr, Cu, Ni and Zn.

**Figure 4 ijerph-19-11374-f004:**
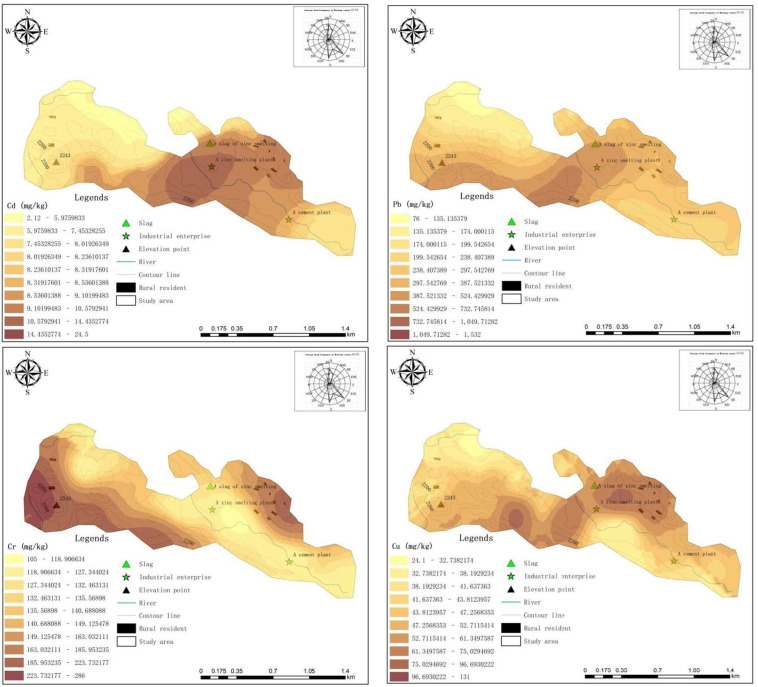
Spatial distribution of Cd, Pb, Cr, Cu, Ni and Zn in surface soil.

**Figure 5 ijerph-19-11374-f005:**
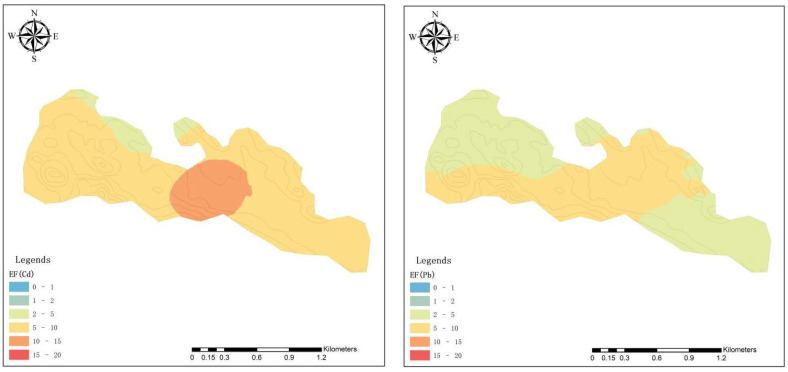
Distribution of enrichment factor (EF) of Cd, Pb, Cr, Cu, Ni and Zn in topsoil.

**Figure 6 ijerph-19-11374-f006:**
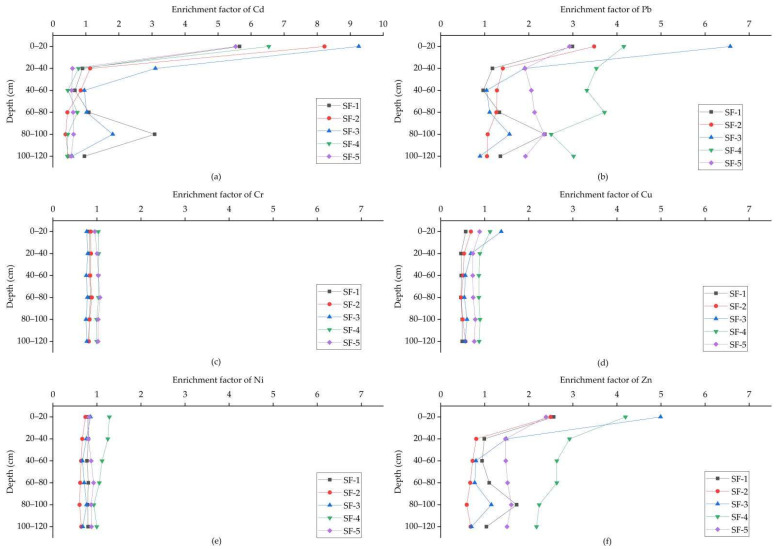
Enrichment factor (EF) of Cd (**a**), Pb (**b**), Cr (**c**), Cu (**d**), Ni (**e**) and Zn (**f**) in soil profiles.

**Figure 7 ijerph-19-11374-f007:**
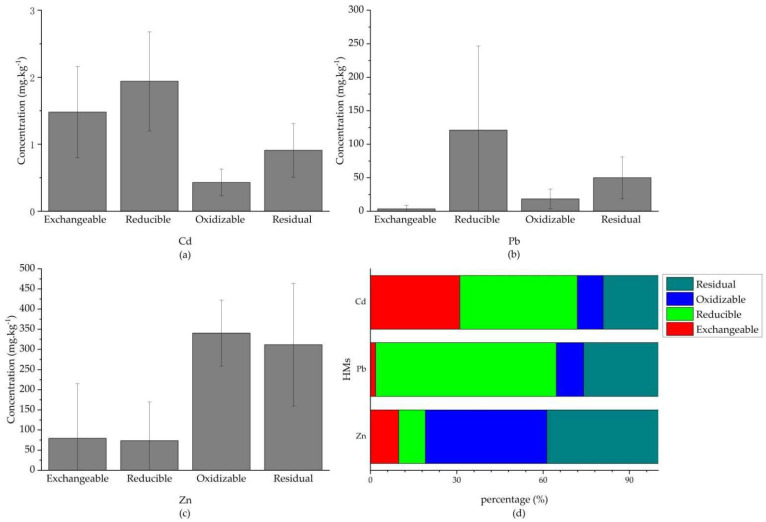
Distribution of chemical fractions for Cd, Pb and Zn (**a**–**c**) and percentages of four chemical fractions for each element (**d**).

**Table 1 ijerph-19-11374-t001:** Classification of enrichment factor (*EF*).

Classification	*EF*	Degree of Enrichment
I	≤1	No enrichment
II	1~2	Slight enrichment
III	2~5	Moderate enrichment
IV	5~20	Significant enrichment
V	20~40	Intense enrichment
VI	>40	Extremely intense enrichment

**Table 2 ijerph-19-11374-t002:** Summary statistics of heavy metals of agricultural soils in study area (mg∙kg^−1^) (n = 107).

Heavy Metals	Min	Max	Mean	SD	CV (%)	Background Value [6]
Cd	2.12	24.5	8.71	3.51	40.3	0.660
Pb	76.0	1530	333	257	77.2	35.2 ± 19.6
Cr	105	286	154	40.0	25.9	95.9 ± 63.2
Cu	24.1	131	51.7	22.1	42.7	32.0 ± 20.8
Ni	32.6	104	61.5	17.5	28.5	39.1 ± 22.4
Zn	210	3340	676	526	77.9	99.5 ± 56.0

SD denotes standard deviation; CV denotes Coefficient of Variation.

**Table 3 ijerph-19-11374-t003:** Concentration of heavy metals in potential sources (mg∙kg^−1^).

Pollution Sources	Cd	Pb	Cr	Cu	Ni	Zn	Sample Size (n)
Irrigation water	1.03 ± 1.02	11.0 ± 12.4	-	-	-	33.9 ± 35.9	7
Livestock manure	8.31 ± 3.61	229 ± 106	39.6 ± 19.0	62.7 ± 54.0	18.84 ± 8.24	668 ± 266.0	6
Pesticide	-	0.31 ± 0.06	2.30 ± 0.97	-	0.14 ± 0.01	6.40 ± 4.58	6
Wet deposition	0.68 ± 0.75	14.8 ± 17.78	50.6 ± 20.2	-	-	63.1 ± 72.7	15
Dry deposition	23.0 ± 12.5	2503 ± 1270	110 ± 33.5	105 ± 22.9	43.4 ± 43.4	3170 ± 2559	15
Rock	1.08 ± 0.87	4.84 ± 4.84	14.1 ± 5.57	3.41 ± 2.52	21.6 ± 10.2	35.4 ± 14.4	7
Fertilizer	0.12 ± 0.05	1.75 ± 1.02	14.8 ± 1.79	3.41 ± 0.45	3.08 ± 0.59	48.3 ± 4.86	2

“-” denotes below detectable limit, and it defaults to zero in CMB model.

**Table 4 ijerph-19-11374-t004:** Calculation results of CMB model.

Species	Calculated	Measured	Irrigation Water	Livestock Manure	Pesticide	Wet Deposition	Dry Deposition	Rock	Fertilizer
Cd	8.710	8.7102	−0.078	0.228	0.000	0.189	0.145	0.514	0.002
Pb	333.133	333.1333	−0.048	0.141	0.001	0.026	0.799	0.080	0.001
Cr	154.287	154.2888	0.000	0.037	0.010	0.682	0.056	0.200	0.015
Cu	51.733	51.7333	0.000	0.813	0.000	0.000	0.151	0.026	0.010
Ni	61.476	61.4755	0.000	0.057	0.001	0.000	0.059	0.875	0.008
Zn	676.384	676.3788	−0.058	0.297	0.006	0.258	0.295	0.190	0.011

R^2^ = 1, χ^2^ = 0.

**Table 5 ijerph-19-11374-t005:** Semivariance models and corresponding correlation coefficients of agricultural soils HMs.

HMs	Fitting Model	Nugget (C_0_)	Sill (C_0_ + C)	C_0_/C_0_ + C (%)	Range (m)	R^2^	RSS
Cd	Spherical	0.0101	0.2552	3.96	2185.0	0.669	0.0343
Pb	Spherical	0.0150	0.3510	4.27	1912.0	0.791	0.0311
Cr	Gaussian	0.0007	0.0514	1.36	1437.6	0.828	6.3990 × 10^−4^
Cu	Gaussian	0.0001	0.1442	0.07	890.3	0.660	0.0105
Ni	Spherical	0.0001	0.0774	0.13	2310.0	0.885	9.9650 × 10^−4^
Zn	Gaussian	0.0035	0.2500	1.4	1312.9	0.723	0.0266

## Data Availability

The datasets of this study are available from the corresponding author on reasonable request.

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
