# Peer review of "Source Identification and Superposition Effect of Heavy Metals (HMs) in Agricultural Soils at a High Geological Background Area of Karst: A Case Study in a Typical Watershed"

_ijerph, 2022, doi:10.3390/ijerph191811374_

Round 1
Reviewer 1 Report
This manuscript researched the source and superposition effect of heavy metals in the soils of a typical karst watershed with high geological background values. In general, this is an interesting work and the results obtained in this study can provide certain meaningful information for local agricultural soil management and remediation. However, a moderate revision is needed to improve the quality of the paper. Some specific comments are listed as follows:
1) Line 48-49, “Therefore, pollution… ecological risk.” Why exogenous sources can enhance the mobility of heavy metals? The mobility of heavy metals seems to be enhanced by the special geological and topographic conditions of karst areas.
2) The significance of identifying source of heavy metals should be explained in the introduction section.
3) Line 88-89, the information of the study site should be more detailed, such as the geographical coordinates and the climatic characteristics.
4) Line 216, Results and discussion?
5) Line 235-238, delete “ According to an …rock weathering”
6) Table 3. The concentrations of heavy metals in the soil samples can’t be 0 mg/kg.
7) Line 246, delete “For further calculating … in this study.”
8) Figure 2-7, the font of the texts in these figures is too small.
9) Line 349-352, delete “According to …Pb and Zn”.
10) Line 356-365, why the distribution characteristics of the chemical fractions of Pb, Cd and Zn were different?
11) The “Conclusion” should be more general and is suggested to be written in a single paragraph.
Reviewer 2 Report
The authors of the article entitled “Source identification and superposition effect of heavy metals (HMs) in agricultural soils at area of geological high background of karst: A case study in a typical watershed" presented research on, sources of HMs in agricultural soil traced by CMB model. Along with the assessing of superposition effect using the spatial analysis of HMs and enrichment factor (EF) and chemical fraction analysis.
In its current form, the article does not meet any requirements, it should be corrected in order for to be accepted for further evaluation.
1. The introduction should be expanded to include research issues and relevant literature, including metal contamination and migration of contaminants in the soil, together with possible threats to the environment. Some literature should also be properly arranged, and not left as a chaotic statement. This would allow for a clearer structure and easier for the reader to understand without having to revisit previously read paragraphs or look for a reference further in the text.
2. Figures in the entire text should not be inserted at the end of the chapter but where there are cited in the text. The scales and legends in all Figures should be standardized. Also the writing style and editing should be according to the MDPI recommendations presented on the IJERPH template, for example line 295 should be corrected.
3. The description of the materials and methods and results should be shortened and clarified. Unfortunately, there are many repetitions in the text that unnecessarily lengthen the description, introducing chaos for the reader.
4. The authors should refer to what other research works and what situations their models can be used. The conclusions should be expanded, especially with the specific values that were received by the authors conducting their research.
5. The language used by the authors is on the verge of colloquial language and not the scientific language that is obligatory for journals!
Additionally, the subject of the article differs from the interests of the International Journal of Environmental Research and Public Health. The suggested Journal would be for example Land in MDPI.
Round 2
Reviewer 2 Report
After reading revised manuscript entitled: “Source identification and superposition effect of heavy metals (HMs) in agricultural soils at area of geological high background of karst: A case study in a typical watershed”, there were mistakes that Authors should correct.
The authors took note of the suggested comments and made the suggested corrections and additions, however, there are still some elements of the manuscript that require attention. After considering all the comments, the article will meet the basic requirements for publication in the journal IJERPH.
1. The introduction was expanded by Authors but still it’s missing more specific information about migration of contaminants in the soil, with possible effects of this threat to the environment. Maybe some suggested literature on a given subject would allow the subject to be fully described.
2. Authors should also ensure that the images or charts inserted into the manuscript are of the best possible quality so that each reader has no problem reading the data, which is influenced by the quality of the graphics and the size of the fonts (e.g. Figure 3 and 6).
3. Formulas used in chapter 2.3. “Chemical mass balance (CMB)” should be centered and without dots and the formula number should be aligned to the right side of the document.
4. In chapter 2.1. should also be photo or map of the research area.
5. Whether the data presented in the Table 2 regarding background values is based on the research of other authors?
6. The main problem is grammar and not necessarily correct wording that has not been fully removed from the text. An example is lines 92, 93, 104 "whoes", 362. Lines 375-376 need to be corrected due to inappropriate beginning of the sentence. Additionally, attention should be paid to the Study site description on lines 105-108.
Additionally, punctuation should be checked throughout the manuscript, which consistently requires a lot of attention due to appearing errors (e.g. lines 93, 275).
Perhaps it is worth for the authors to consider the use of the services of professional English Editor, which would help in unifying and correcting grammar and language, which would certainly improve the quality of the manuscript. Such services are offered, among others, by MDPI.
Suggested literature:
Wang, L.K.; Veysel, E.; Ferruh, E. Handbook of Advanced Industrial and Hazardous Wastes Treatment; CRC Press, Taylor & Francis Group: Boca Raton, FL, USA, 2009.
Tack, F.M.G.; Bardos, P. Overview of Soil and Groundwater Remediation. In Soil and Groundwater Remediation Technologies; Ok, Y.S., Rinklebe, J., Hou, D., Tsang, D.C.W., Tack, F.M.G., Eds.; Taylor & Francis: Oxfordshire, UK, 2020.
Pusz, A.; WiÅ›niewska, M.; Rogalski, D. Assessment of the Accumulation Ability of Festuca rubra L. and Alyssum saxatile L. Tested on Soils Contaminated with Zn, Cd, Ni, Pb, Cr, and Cu. Resources, 2021, 10 (5), 1–18.
Saha, J.K.; Selladurai, R.; Coumar, M.V.; Dotaniya, M.L.; Kundu, S.; Patra, A.K. Assessment of Heavy Metals Contamination in Soil. In Soil Pollution—An Emerging Threat to Agriculture; Environmental Chemistry for a Sustainable World; Springer: Singapore, 2017; Volume 10.
